# High prevalence of injection drug use and blood-borne viral infections among patients in an urban emergency department

**Erik S. Anderson**❓*, **Carly Russell, Kellie Basham, Martha Montgomery, Helen Lozier**❓, **Abigail Crocker, Marisa Zuluaga, Douglas A. E. White**

Department of Emergency Medicine, Alameda Health System, Highland Hospital, Oakland, CA, United States of America

\* esanderson@alamedahealthsystem.org

## Abstract

### Background

The opioid epidemic has led to an increase in the number of persons who inject drugs, and this population accounts for 12% of new human immunodeficiency virus (HIV) and 60% of new hepatitis C virus (HCV) infections in the United States annually. While persons who inject drugs disproportionately utilize the emergency department (ED), accurate data is lacking on the prevalence and patterns of injection drug use, and prevalence of co-occurring HIV and HCV infections among ED patients.

### Objective

The primary outcome was to assess the prevalence of injection drug use and co-occurring HIV and HCV infection among patients presenting to an urban ED.

### Methods

This was a cross sectional study conducted at an urban ED, with an annual census of 65,000 visits. A closed-response questionnaire was developed based on publicly available validated surveys to assess patterns of injection drug use and HIV and HCV infection status, and administered by trained research assistants to all registered adult patients during 4-hour blocks of time.

### Results

Of the 2,319 eligible patients, 2,200 (94.9%) consented and completed the survey. 241 (11.0%) had ever used injection drugs, 103 (4.7%) currently used injection drugs, and 138 (6.3%) formerly used injection drugs. White patients age 25 to 34 years and white patients age 55 to 64 years had the highest prevalence of current (25.6%) and former (27.1%) injection drug use, respectively. Persons who use injection drugs had a higher prevalence of HCV infection (52.7% vs. 3.4%) and HIV infection (6.2% vs. 1.8%) than the rest of the population.

**Data Availability Statement:** The data underlying the results presented in the study are available from Alameda Health System Institutional Review

Board after appropriate review and patient information is de-identified. The contact person for such queries is Jen Sun, jennsun@alamedahealthsystem.org

**Funding:** Authors ESA, CR, KB, HL, AC, and DAEW, received funding from Gilead Sciences FOCUS Grant. The funders had no role in study design, data collection and analysis, decision to publish, or preparation of the manuscript.

**Competing interests:** We have read the journal's policy and the authors of this manuscript have the following competing interests: ESA, CR, KB, HL, AC, and DAEW received funding from Gilead Sciences FOCUS Grant. The grant provided partial salary support and program support for HIV and HCV screening in the emergency department. This does not alter our adherence to PLOS ONE policies on sharing data and materials.

## Conclusion

A high prevalence of ED patients report injection drug use, and this population self-reports a high prevalence of HIV and HCV infection. Emergency departments are in a unique position to engage with this population with regards to substance use treatment and linkage to care for HIV and HCV infection.

## Introduction

The opioid epidemic has led to an increase in the number of persons who inject drugs (PWID). Most notably, this increase in PWID has been related to the opioid epidemic and non-medical use of prescription opioids [1]. The high-risk practice of sharing needles, syringes, and other drug injection equipment put PWID at high risk for acquiring and transmitting human immunodeficiency virus (HIV) and hepatitis C virus (HCV) infection. Additionally, PWID are exposed to significant adverse risk environments that can lead to HIV and HCV infections, including homelessness, high-risk sexual practices, and incarceration [2,3]. PWID account for 12% of new HIV and 60% of new HCV infections in the United States (US) annually, and there has been a marked increase in HCV transmission directly related to the opioid epidemic and PWID [4,5].

In addition to blood-borne viral infections, PWID have a complex set of health-related problems, including high rates of skin and soft tissue infections, overdose, and homelessness, and low rates of health insurance and access to primary care [6–12] which contribute to disproportionate emergency department (ED) utilization [8,9]. As a result, some EDs have implemented public health interventions and treatment for PWID, including routine HIV and HCV screening, substance use treatment, and other harm reduction interventions [12–15].

Unfortunately, accurate data is lacking on the number of PWID in the ED, and available studies use probability surveys to estimate utilization or a combination of international classification disease codes to estimate PWID prevalence in various care settings [4,16,17]. Understanding the prevalence of injection drug use, patterns of use, and prevalence of co-occurring HIV and HCV infections can help EDs develop, refine, and disseminate these programs aimed to mitigate the harms associated with injection drug use.

The main goals of this study were to estimate the PWID prevalence among patients presenting to an urban ED and to compare the HIV and HCV co-infection rates between those who report injection drug use and those who do not.

## Materials and methods

### Study design

We performed a cross-sectional study of adult patients presenting to the Highland Hospital ED. This study was approved by the Highland Hospital, Alameda Health System (AHS), institutional review board. Patients provided written informed consent to participate.

### Study setting and population

Highland Hospital is an urban teaching hospital in Oakland, California with an accredited 4-year Emergency Medicine residency program. The annual ED volume is approximately 65,000 patients; 2% of patients are less than 12 years of age; approximately 40% of patients are black, 40% Hispanic, 15% white; and 45% are female. Roughly 75% of patients have public

insurance, 15% are uninsured, and 10% of patients have private insurance. The Highland Hospital ED has a formal program for the initiation of buprenorphine for patients with opioid use disorder, as well as harm reduction initiatives for patients who use injection drugs, and provides routine HIV and HCV screening to adult patients. Approximately 25% of all adult ED patients have received annual HIV and HCV screening since 2014. Additionally, community clinics and syringe services programs have offered routine HIV and HCV screening for PWID, resulting in relatively high rates of knowledge about blood-borne viral infections in this patient population.

## Selection of participants

Adult patients ≥18 years were eligible for survey administration if they completed triage registration, spoke Spanish or English, were medically stable, and able to provide informed consent. Surveys were conducted in private areas in the waiting room, Fast Track, or the main ED.

## Survey content and administration

The survey was developed by study staff with expertise in this field (DAEW and ESA). The survey elicited patient injection drug use history and self-report of HIV and HCV infection and treatment (S1 Appendix). To assess personal history of injection drug use and frequency of use, we used validated questions from the National Institute on Drug Abuse Clinical Trials Network-recommended Common Data Elements (CDE) of Substance Use Disorders for use in clinical trials and electronic health records (EHR) (CDE: 3269978 and CDE:3269986, respectively) [18]. The survey was pilot tested on a convenience sample of 15 patients and revised to ensure content validity and patient comprehension.

Five volunteer research assistants (RAs), who were blinded to the study purpose, administered the surveys from November 2018 through April 2019. The RAs worked in pairs during assigned 4-hour time blocks on pre-set weekdays from 10am to 2pm. At the beginning of each shift, RAs reviewed the ED tracking board to identify all adult patients who were registered, which represented the cohort of potentially eligible patients. Patients were deemed medically stable if they were awake and alert and able to participate in conversation with the RA. In an effort to prevent selection bias, RAs were blinded to any test results, the reason for the patient's visit, and were not given access to the patient's medical record. Further study eligibility was evaluated at the bedside and participating patients completed written consent prior to survey administration. Participants were not compensated for participation in the survey. Patients were able to decline a survey if they reported participating in the past.

## Outcome measures

The primary outcome was the proportion of surveyed patients who had ever used injection drugs. Secondary outcomes were the proportion of surveyed patients with HIV or HCV co-infection and ED utilization. Emergency department utilization was defined as the number of ED visits during the year prior to the index visit.

## Data analysis

All patients who completed the survey were analyzed. Any duplicate patients were removed, and only the initial survey results were used as part of the analysis. Descriptive analyses were performed for all variables. Categorical data are reported as numbers and percentages and continuous data are reported as means with standard deviation (SD). When appropriate, absolute differences in proportion with 95% confidence intervals (CI) were calculated, and odds ratios

were calculated to display relative differences between categories. Statistical analyses were performed using Microsoft Excel (Version 14.3.7, Microsoft Corporation) and Stata (Version 13, Stata Corporation, College Station, TX). Injection drug use was defined as any non-medically sanctioned drug taken by intravenous, intramuscular, or subcutaneous routes for non-medical use, and included opioids, amphetamine-type stimulants, and cocaine. Based on their self-reported injection drug use, surveyed patients were categorized as either *never* PWID or *ever* PWID. The latter group was further dichotomized as *former* PWID (defined as people who have not injected drugs in the previous 12 months) or *current* PWID (defined as people who have injected drugs in the previous 12 months). The clinical rational for this distinction was current PWID are felt to be at the greatest risk of transmitting and acquiring blood-borne viral infections through injection drug use practices while former PWID are assumed to have permanently ceased injection drug use and are therefore not at increased risk for viral disease transmission.

## Results

Between November 2018 and April 2019 there were a total of 24,309 patients who presented to the ED for care, of whom 4,334 (17.8%) were registered during study hours when RAs were enrolling patients. Of the patients who were registered during study hours, 2,319 (53.5%) were eligible for survey administration and 2,200 consented and completed the survey. Fig 1 shows the flow of eligibility and assessment for study participation, and reasons that patients were ineligible for survey administration. Of the 2,015 patients who were registered in the ED during hours RAs were present but not surveyed, 877 (43.5%) were discharged before the RA was able to meet the patient at the bedside, 653 (32.4%) were unavailable due to ongoing patient care, 447 were previously surveyed (22.2%), 415 had altered mental status (20.6%), 259 were critically ill (12.9%), and 32 had language barriers (1.6%).

Characteristics of the unique ED census and surveyed patients, stratified by injection drug use history, can be found in Table 1. The mean age was 49.8 (SD 16.4), 964 (43.8%) were female, 909 (41.3%) black, 678 (30.8%) Hispanic, and 311 (14.1%) were white. Of surveyed patients, 241 (11.0%; 95% CI: 9.7% to 12.3%) were ever PWID, 103 (4.7%; 95% CI: 3.8% to 5.6%) were current PWID, and 138 (6.3%; 95% CI: 5.3% to 7.4%) were former PWID. Compared to the cohort of never PWID, ever PWID were more likely to be male (73.4% vs 56.2%; difference 17.2%; 95% CI: 11.2% to 23.2%) and had higher rates of homelessness (16.2% vs 4.8%; difference 11.4%; 95% CI: 6.7% to 16.1%). Current compared to former PWID were similar demographically, with the exception of age: the average age of current PWID was 45.0 (SD 13.3) and the average age of former PWID was 56.7 (SD 13.4) (p<0.001).

Although hospital admission rates were similar between cohorts, ever PWID visited the ED more often in the preceding 12 months than never PWID (ever PWID: median 2 visits, IQR 1 to 4; never PWID: median 1 visit, IQR 1 to 2; p<0.001). There was no difference in the median number of ED visits or admission rates between former and current PWID.

Fig 2A and 2B compare the prevalence of current and former injection drug use stratified by age group and race. White patients aged 25 to 34 years had the highest prevalence of current injection drug use (25.6%, 95% CI: 13.5% to 41.2%) and white patients aged 55 to 64 years had the highest prevalence of former injection drug use (27.1%, 95% CI: 17.2% to 39.1%).

The prevalence of self-reported HCV and HIV infection among surveyed patients can be found in Table 2. The ever PWID cohort had a higher prevalence of HCV infection (52.7% vs. 3.4%; difference 49.3%; 95% CI: 42.9% to 55.6%) and HIV infection (6.2% vs. 1.8%; difference 4.4%, 95% CI: 1.3% to 7.5%) than never PWID. The prevalence of HCV infection was similar between current and former PWID (45.6% vs 58.0%; difference 12.3%; 95% CI: -0.3% to

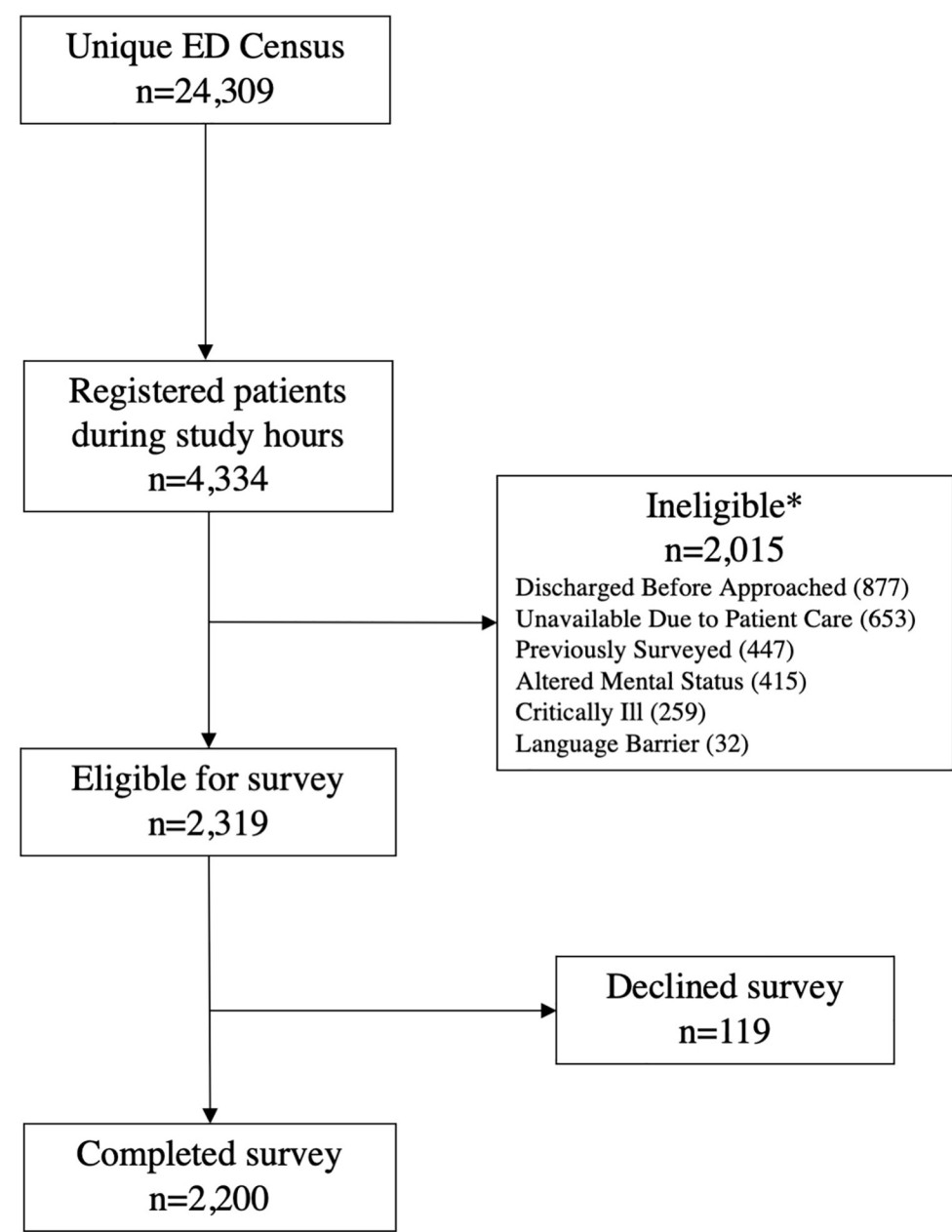

*Several patients had more than one reason for ineligibility recorded

**Fig 1. Details of study enrollment and survey completion at Highland Hospital from November 2018 through April 2019.**

25.0%), as was the prevalence of HIV infection (5.8% vs 6.5%; difference 0.7%; 95% CI: -5.4% to 6.8%), although there was a trend toward slightly higher prevalence in former PWID.

The reported continuums of care for HIV and HCV-infected patients can be found in Table 3. 29.4% of all HIV infections and 65.4% of all HCV infections are among the ever

**Table 1. Characteristics and results of surveyed Highland Hospital emergency department patients from November 2018 through April 2019.**

| | Unique Census | All Patients | Never PWID[a] | Ever PWID[b] | Absolute Difference % | Unadjusted Odds Ratio |
|---|---|---|---|---|---|---|
| | N = 24,309 | N = 2,200 (%) | N = 1,959 (%) | N = 241 (%) | (95% CI) | |
| **Age,** mean (SD), y | 44.6 (17.0) | 49.8 (16.4) | 49.5 (16.6) | 51.7 (14.5) | 2.2 (0.0 to 4.4) | – |
| Age 18–24 | 2,931 (12.1) | 136 (6.2) | 133 (6.8) | 3 (1.2) | -5.6 (-7.4 to -3.8) | 0.2 (0.1 to 0.5) |
| Age 25–34 | 5,572 (22.9) | 366 (16.6) | 326 (16.6) | 40 (16.6) | 0.0 (-5.0 to 5.0) | 1.0 (0.7 to 1.4) |
| Age 35–44 | 4,788 (19.7) | 389 (17.7) | 345 (17.6) | 44 (18.3) | 0.7 (-5.9 to 4.5) | 1 (0.7 to 1.5) |
| Age 45–54 | 4,133 (17.0) | 398 (18.1) | 365 (18.6) | 33 (13.7) | -4.9 (-9.6 to -0.2) | 0.7 (0.5 to 1.0) |
| Age 55–64 | 3,753 (15.4) | 494 (22.5) | 422 (21.5) | 72 (29.9) | 8.4 (2.3 to 14.5) | 1.6 (1.2 to 2.1) |
| Age 65–74 | 1,955 (8.0) | 276 (12.5) | 231 (11.8) | 45 (18.7) | 6.9 (1.8 to 12.0) | 1.7 (1.2 to 2.4) |
| Age 75+ | 1,177 (4.8) | 133 (6.0) | 129 (6.6) | 4 (1.7) | -4.9 (-6.8 to -2.9) | 0.2 (0.1 to 0.6) |
| **Race/Ethnicity**[c] | | | | | | |
| Black | 8,876 (38.7) | 909 (41.3) | 793 (40.5) | 116 (48.1) | 7.6 (0.9 to 14.2) | 1.4 (1.0 to 1.8) |
| Hispanic/Latino | 8,472 (40.0) | 678 (30.8) | 638 (32.6) | 40 (16.6) | -16.0 (-21.1 to -10.9) | 0.4 (0.3 to 0.6) |
| White | 3,183 (13.9) | 311 (14.1) | 238 (12.1) | 73 (30.3) | 18.2 (12.2 to 24.2) | 3.1 (2.3 to 4.3) |
| Asian | 2,150 (9.4) | 190 (8.6) | 184 (9.4) | 6 (2.5) | -6.9 (-9.3 to -4.5) | 0.2 (0.1 to 0.6) |
| Other | 239 (1.04) | 23 (1.0) | 21 (1.1) | 2 (0.8) | -0.3 (-1.5 to 0.9) | 0.8 (0.2 to 3.3) |
| **Gender** | | | | | | |
| Female | 11,162 (45.9) | 964 (43.8) | 900 (45.9) | 64 (26.6) | -19.3 (-25.3 to -13.3) | 0.4 (0.3 to 0.6) |
| Male | 13,146 (54.1) | 1,236 (56.2) | 1,059 (54.1) | 177 (73.4) | 19.3 (13.3 to 25.3) | 2.4 (1.7 to 3.2) |
| **Insurance** | | | | | | |
| Medicaid | 14,880 (61.5) | 1,411 (64.1) | 1,255 (64.1) | 156 (64.7) | 0.6 (-5.8 to 7.0) | 1.0 (0.8 to 1.4) |
| Medicare | 3,194 (13.2) | 439 (20.0) | 372 (19.0) | 67 (27.8) | 8.8 (2.9 to 14.7) | 1.6 (1.2 to 2.2) |
| Uninsured | 3,587 (14.8) | 179 (8.1) | 173 (8.8) | 6 (2.5) | -6.3 (-8.6 to 4.0) | 0.5 (0.2 to 0.9) |
| Private | 2,475 (10.2) | 161 (7.3) | 152 (7.8) | 9 (3.7) | -4.1 (-6.7 to -1.4) | 0.5 (0.2 to 0.9) |
| **Homeless** | 763 (3.1) | 106 (4.8) | 67 (3.4) | 39 (16.2) | 12.8 (8.1 to 17.5) | 5.5 (3.6 to 8.3) |
| **Disposition** | | | | | | |
| Admitted | 3,506 (14.4) | 655 (29.8) | 578 (29.5) | 77 (32.0) | 2.5 (-3.7 to 8.7) | 1.1 (0.8 to 1.5) |
| Discharged | 20,803 (85.6) | 1,545 (70.2) | 1,381 (70.5) | 164 (68.0) | -2.5 (-8.7 to 3.7) | 0.9 (0.7 to 1.2) |
| **ED Visits during study period**, median (IQR) | 1 (1 to 1) | 1 (1 to 3) | 1 (1 to 2) | 2 (1 to 4) | P<0.014 | – |

PWID, person who uses injection drugs; CI, confidence interval; SD, standard deviation; Y, years; ED, emergency department; IQR, interquartile range.

[a]Surveyed patients who report never injecting drugs.

[b]Surveyed patients who report ever injecting drugs, includes current and former injection drug use.

[c]Ethnicity is non-Hispanic unless noted.

Absolute differences and odds ratios are calculated as a comparison between the Never PWID and Ever PWID cohorts. Comparison of median ED visits using Wilcoxon rank sum test.

PWID cohort. Rates of being in HIV care (77.8% vs. 61.1%) and taking antiretroviral therapy (ART) (77.8% vs. 52.8%) were similar for former and never PWID. While not statistically significant, current PWID trended towards having lower rates of being in HIV care (50%) and on ART (33.3%) compared to former PWID. Similarly, rates of being in HCV care (58.8% vs. 55.2%) and achieving HCV cure (45.0% vs. 37.3%) were similar for former and never PWID, while current PWID had lower rates of being in HCV care (14.9%) and achieving HCV cure (8.5%) compared to either group (p<0.001).

## Limitations

This study is limited by its single center design. Regional variation of the prevalence of PWID can be significant which limits the generalizability of our findings. Our ED treatment

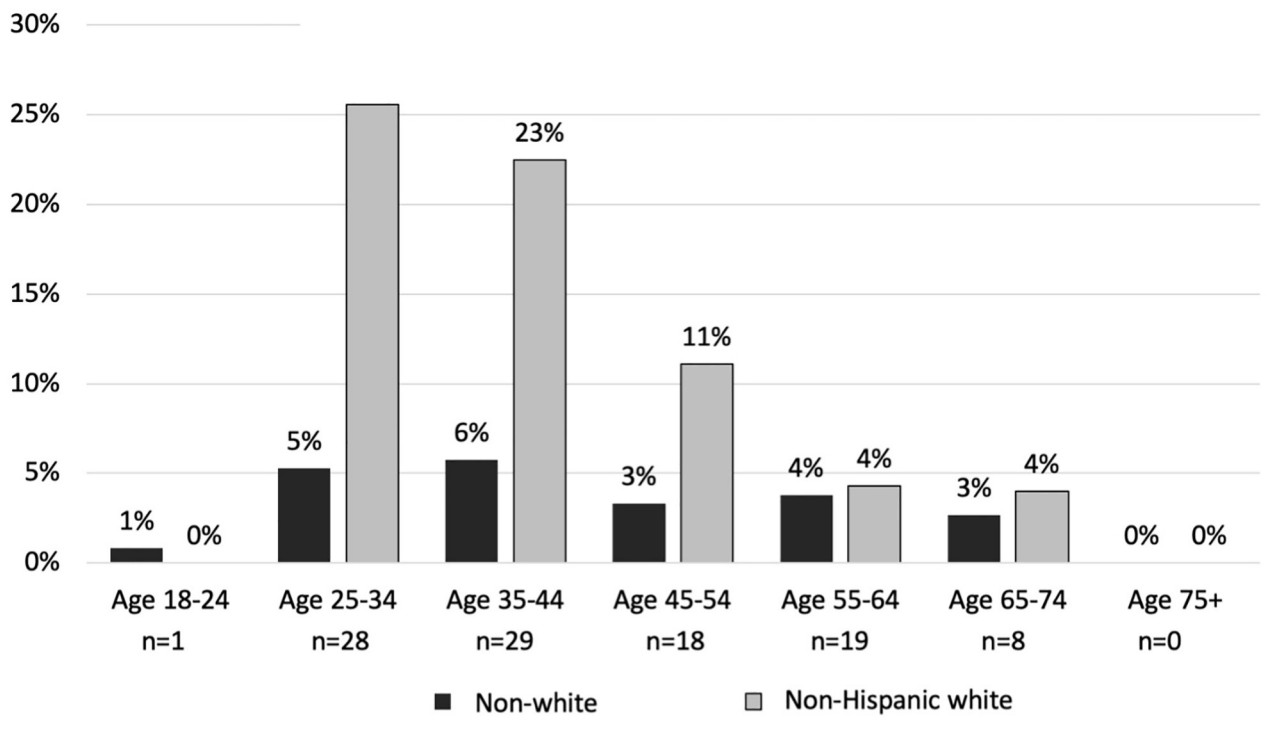

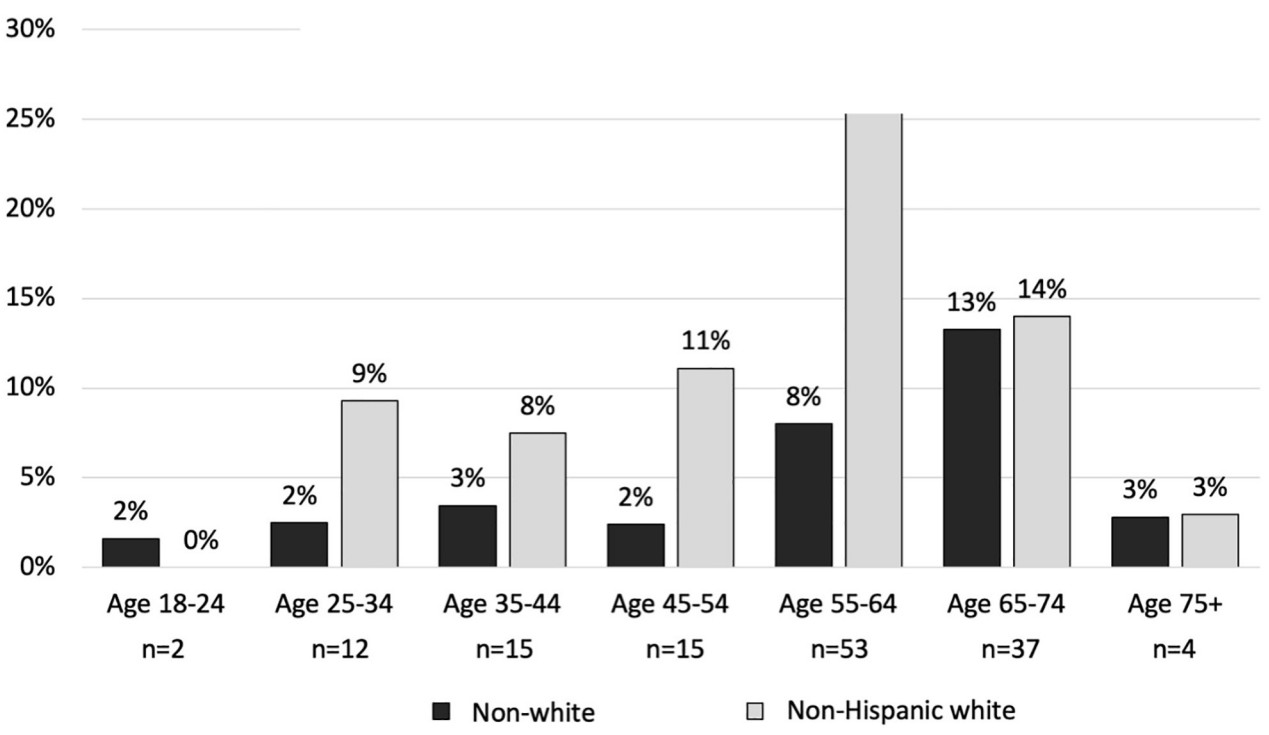

**Fig 2.** (a) Prevalence of current injection drug use by age and race among 2,200 surveyed emergency department patients. (b) Prevalence of former injection drug use by age and race among 2,200 surveyed emergency department patients. Legend: Non-white includes Hispanic/Latino, Black, Asian, and other race and ethnicity categories. Current injection drug use indicates surveyed patients who report injecting drugs in the previous 12 months; former injection drug use indicates surveyed patients who report injecting drugs, but not in the past 12 months. Prevalence is based on the total number of patients surveyed in each age group, and 'n' indicates the number of patients that report current or former injection drug use.

programs for opioid use disorder and harm reduction initiatives may actually skew our results by promoting increased ED utilization by PWID compared to other EDs that do not provide such services.

Additionally, the stigma associated with injection drug use, HIV, and HCV infection may compromise the validity of patient responses, possibly leading to underreporting of drug use, co-infection with HIV and HCV infection, and inaccurate assessments of current patterns of use and treatment adherence. Further, we dichotomized ever PWID into two groups, former and current PWID, in an attempt to identify the subgroup of active users who have with the greatest risk of transmitting and acquiring HIV and HCV infection and therefore are of the most public health concern [19]. Given the relapsing and remitting nature of injection drug use the assumption that all former PWID have permanently ceased using and may not reflect clinical reality.

We also note that due to the convenience sample methodology, it is unclear the impact of selection bias on our results. We report our study sample alongside the information about our total ED census, which helps to provide context to our results. Our findings rely on patient self-report, which, although accurately represents patient knowledge about their HIV and HCV infection status, may not reflect the true seroprevalence of disease in each cohort. Lastly, the low absolute numbers of PWID HIV and HCV-infected patients included in this study prevents making a detailed comparative analysis.

## Discussion

In this survey study of 2,200 patients in a Northern California urban ED, we found a significant proportion were ever PWID (11%), of which 45% were current PWID. Additionally, we report a significantly higher prevalence of HIV and HCV co-infection among ever PWID (HIV 6.2%; HCV 53%) than never PWID (HIV 1.8%; HCV 3.4%). Lastly, ED patients with HIV or HCV infection who were also current PWID were less likely to receive care for their infections than former and never PWID.

Injection drug use is a significant driver of HCV and HIV transmission through high-risk practices of sharing needles, syringes, and other drug injection equipment, as well as from associated risk environments [2,3]. A focus on PWID and efforts to address substance use,

**Table 2. Prevalence of self-reported HIV and HCV infection among surveyed emergency department patients stratified by injection drug use history.**

|  | All | Never PWID[a] | Ever PWID[b] | Current PWID[c] | Former PWID[d] |
|---|---|---|---|---|---|
|  | N = 2,200 (%) | N = 1,959 (%) | N = 241(%) | N = 103 (%) | N = 138 (%) |
| HIV Diagnosis | 51 (2.3) | 36 (1.8) | 15 (6.2) | 6 (5.8) | 9 (6.5) |
| HCV Diagnosis | 194 (8.8) | 67 (3.4) | 127 (52.7) | 47 (45.6) | 80 (58.0) |

*HIV*, human immunodeficiency virus; *HCV*, hepatitis C virus; *PWID*, *person who uses injection drugs*.

[a]Surveyed patients who report never injecting drugs.

[b]Surveyed patients who report ever injecting drugs, includes current and former injection drug use.

[c]Surveyed patients who report injecting drugs in the previous 12 months.

[d]Surveyed patients who report injecting drugs, but not in the past 12 months.

**Table 3. Continuum of care for emergency department with self-reported HIV or HCV infection, stratified by injection drug use history.**

|  | Never PWID[a] | Ever PWID[b] | Current PWID[c] | Former PWID[d] |
|---|---|---|---|---|
| HIV Diagnosis | 36 | 15 | 6 | 9 |
| In Care (%) | 22 (61.1) | 10 (66.7) | 3 (50) | 7 (77.8) |
| On ART (%) | 19 (52.8) | 9 (60.0) | 2 (33.3) | 7 (77.8) |
| HCV Diagnosis | 67 | 127 | 47 | 80 |
| In Care (%) | 37 (55.2) | 54 (42.5) | 7 (14.9) | 47 (58.8) |
| Cured (%) | 25 (37.3) | 40 (31.5) | 4 (3.9) | 36 (45.0) |

*HIV*, human immunodeficiency virus; *ART*, antiretroviral treatment; *HCV*, hepatitis C virus; *PWID*, person who uses injection drugs.

[a]Surveyed patients who report never injecting drugs.

[b]Surveyed patients who report ever injecting drugs, includes current and former injection drug use.

[c]Surveyed patients who report injecting drugs in the previous 12 months.

[d]Surveyed patients who report injecting drugs, but not in the past 12 months.

therefore, are important steps to reduce the incidence of new HIV and HCV infections. In fact, a history of injection drug use is the most important risk factor for new HCV infection, and a focus on treatment as prevention for PWIDs is critically important. Mathematical models estimate that if only 1% to 7% of PWID with chronic HCV infection are treated, the prevalence of disease would be halved in 15 years [19,20]. Likewise, PWID are 22 times more likely to acquire HIV than the rest of population, and HIV can spread rapidly through communities of PWID [21]. In a 2015 HIV outbreak among 181 people in Scott County, Indiana, approximately 90% of infections were associated with injection of prescriptions opioids, and over 90% of patients were co-infected with HCV [22].

Our results are in line with estimates of PWID prevalence among urban ED populations in other parts of the US. An HCV seroprevalence study in the Johns Hopkins ED in Baltimore, Maryland, found that 7% of the patients in their cohort were ever PWID based on chart review [23]. Another study of over 22,000 patients from 4 geographically distinct US EDs found that 8% of ED patients responded "yes" to ever using injection drugs; the study utilized the Denver Health Risk Score for targeted HIV screening [24]. (Jason Haukoos, personal communication). Our findings, taken in context with prior ED cross sectional data, suggest that the prevalence of PWID among urban ED patients may lie between 7% and 11%. This represents a significant increase compared with estimates from earlier in the opioid epidemic. A report using a nationwide sample of respondents who received ED care between 2007–2009 found only 2.4% of respondents had a history of injection drug use [25].

We also found a high prevalence of both HIV infection (6.2%) and HCV infection (53%) among PWID in our urban ED, both several-fold higher than patients who reported never using injection drugs. Moreover, the true prevalence of these infections in our PWID cohort is likely even higher than these estimates based on self-report. We previously found that many of the current PWID in our ED had not undergone recent HIV or HCV screening. Our findings underscore the importance of protocolized HIV and HCV screening for all ED patients who are identified as PWID.

The ED is at the nexus of care for PWID and those with HIV and HCV infections [9]. We believe a multifaceted, ED-based public health approach is needed, which includes strategies promoting access to clean needles and safe injection sites; routine screening for HIV and HCV infection; easy access to medications for opioid use disorder; streamlined treatment pathways for HIV and HCV infection and substance use disorders; and co-localization of care and treatment resources for substance use and blood-borne viral infections. Some EDs, including ours,

have implemented programs that initiate treatment for patients with opioid use disorder, including a dedicated, low-barrier linkage to care pathway to an addiction medicine clinic. In addition, patients identified with HIV or HCV infection through our longstanding ED-based screening program are also screened for co-occurring substance use disorders and referred to the substance use pathway.

Despite these interventions, engagement and linkage to care for ED PWID remains challenging.

We found large drop-offs in the linkage to care and treatment rates for surveyed ED patients with HIV and HCV infection, with the lowest rates among PWID. National estimates report that nearly 80% of PWID living with HIV are linked to care, but only 50% are retained in care [26]. Tsui et al. found that among a cohort of PWID in Seattle, only 17% had received ART for their HCV infection and less than 10% being were cured [27]. While it is recognized that some of the barriers to care faced by PWID are directly related to substance use disorders, additional factors such as lack of transportation, unstable housing, ineffective treatment for concomitant, non-opioid related substance use disorders, and stigma and bias encountered within the healthcare system, are likely contributory.

While comprehensively addressing the intertwined public health issues of injection drug use and HIV and HCV infection has improved care and treatment for some patients, the majority have complex needs and face significant barriers to care. Given the significant impact of these illnesses, and the central role that the ED plays in the care of these patients, further ED research aimed at this high-risk cohort is needed, evaluating best practices, as well as new and creative initiatives.

## Conclusion

A high prevalence of ED patients are PWID, and this population self-reports a high prevalence of HIV and HCV infection. Emergency departments are in a unique position to engage with this population with regards to substance use treatment as well as screening and engagement to care for HIV and HCV infection.

## Supporting information

**S1 Appendix.**
(PDF)

## Acknowledgments

The authors would like to acknowledge Bradley Frazee, MD his support in this project.

## Author Contributions

**Conceptualization:** Erik S. Anderson, Douglas A. E. White.

**Data curation:** Erik S. Anderson, Carly Russell, Kellie Basham, Martha Montgomery, Helen Lozier, Abigail Crocker, Marisa Zuluaga.

**Formal analysis:** Erik S. Anderson.

**Funding acquisition:** Erik S. Anderson, Douglas A. E. White.

**Investigation:** Erik S. Anderson, Carly Russell, Kellie Basham, Martha Montgomery, Helen Lozier, Abigail Crocker, Marisa Zuluaga, Douglas A. E. White.

**Methodology:** Erik S. Anderson, Douglas A. E. White.

**Project administration:** Erik S. Anderson, Kellie Basham, Douglas A. E. White.

**Supervision:** Erik S. Anderson, Carly Russell, Douglas A. E. White.

**Writing – original draft:** Erik S. Anderson, Martha Montgomery, Douglas A. E. White.

**Writing – review & editing:** Erik S. Anderson, Douglas A. E. White.

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
