## [Decision Letter · Decision Letter 0]

25 Mar 2020

PONE-D-20-02701

High Prevalence Of Injection Drug Use And Blood-Borne Viral Infections Among Patients In An Urban Emergency Department

PLOS ONE

Dear Dr. Anderson,

Thank you for submitting your manuscript to PLOS ONE. After careful consideration, we feel that it has merit but does not fully meet PLOS ONE’s publication criteria as it currently stands. Therefore, we invite you to submit a revised version of the manuscript that addresses the points raised during the review process.

The authors should address all of the critiques raised by the review, and which I concur with. I also urge the authors to be more transparent in the abstract regarding prevalence of self-reported injection drug use the the infections HIV and HCV. Information about those not surveyed should be provided for comparison purposes if possible. Identifying strengths that can support the potential impact of the study would add to the manuscript. 

We would appreciate receiving your revised manuscript by April 25, 2020. To enhance the reproducibility of your results, we recommend that if applicable you deposit your laboratory protocols in protocols.io, where a protocol can be assigned its own identifier (DOI) such that it can be cited independently in the future. For instructions see: http://journals.plos.org/plosone/s/submission-guidelines#loc-laboratory-protocols

We look forward to receiving your revised manuscript.

Kind regards,

Kimberly Page, PhD, MPH

Academic Editor

PLOS ONE

Additional Editor Comments (if provided):

I concur I concur with the review regarding the weaknesses of the study: including the lack of information about people who did not take the survey, the survey instrument validity, the weaknesses of self-reported infection status and editing the figures. However, I think that the authors can address these critiques with a major revision, noting the more carefully the strengths and weaknesses of the study. For example what are HIV and HCV testing rates in this area? If they are high, do they inspire confidence in the self-reported prevalence. Addressing the potential biases of the study are important and not over-interpreting the results.

Journal Requirements:

2. Please ensure that you refer to Figure 1 in your text as, if accepted, production will need this reference to link the reader to the figure.

"I have read the journal's policy and the authors of this manuscript have the following competing interests:

ESA, CR, KB, HL, AC, and DAEW received funding from Gilead Sciences FOCUS Grant. The grant provided partial salary support and program support for HIV and HCV screening in the emergency department. "

Reviewers' comments:

Reviewer's Responses to Questions

**Comments to the Author**

1. Is the manuscript technically sound, and do the data support the conclusions?

Reviewer #1: Partly

2. Has the statistical analysis been performed appropriately and rigorously? 

Reviewer #1: No

3. Have the authors made all data underlying the findings in their manuscript fully available?

Reviewer #1: No

4. Is the manuscript presented in an intelligible fashion and written in standard English?

Reviewer #1: Yes

5. Review Comments to the Author

Reviewer #1: The manuscript details results from a convenience sample survey of approximately 2,200 emergency department (ED) patients at a single urban ED (Highland) in Oakland, CA. The survey focuses on assessing injection drug use (IDU) history and self-reported HIV and hepatitis C (HCV) infections. The authors report high rates of IDU (and HIV and HCV) among their sample. The findings are relatively limited, including only a few demographic and visit characteristics. Within their findings, the note high reported IDU (e.g., 26% in respondents 25-34 years) and generally high rates of HIV (6%) and HCV (53%) infection.

The authors report a very high response rate--about 95%. The challenge is interpreting this number is that the authors do not describe the characteristics of 2,015 patients that they did not approach--and critically why not? Where these patients too ill? Intoxicated? Spoke a language other than English or Spanish? Discharged before the RA could approach them for participation? This information is critical to judge the validity of the remaining responses. The convenience sample design is practical, but it may have important implications on the findings. For example, if patients who inject drugs (or have HIV or HCV infection) have longer ED visits lengths of stay compared to their null counterparts, then the findings are subject to length biased sampling--and the results may be biased.

The convenience sample design, single site selection, and lack of methodologic details limit the general applicability of their findings.

Specific comments:

1. Abstract: The authors state that they used a 'validated questionnaire' in the abstract. This is a bit of an overstatement. The authors used 2 validated question items from the NIDA Clinical Trials Network (CTN) Common Data Elements (CDE). This is a strength in their methods, but a general overstatement to imply that the questionnaire was validated (it would be fair to say that 2 items were validated). The authors do not provide their questionnaire for review. They do not describe the other survey data elements.

2. The authors should clarify their use of 'Hispanic,' 'white,' and 'black'. In many contexts, Hispanic refers to ethnicity and white and black refer to race. I would presume that 'white' means 'non-Hispanic white' and 'black' means 'non-Hispanic black'. Again, if the questionnaire were provided as supplemental material, it would help gauge how the questions were asked.

3. Were subjects compensated in any way? Were there any duplicate respondents (more than 1 ED visit during the study period)? If so, how were they managed?

4. The statistical analysis is relatively simple, reporting percentages and absolute percentage differences between categories. The manuscript would be strengthened by a more robust analysis that included both absolute and relative differences (whether reporting as prevalence ratios or odds ratios). There are limited attempts to detect confounding or interaction (via limited stratification provided by text and not shown in tables). Sample size restrictions (of reported IDU, HIV, and HCV) limit the ability to adjust for confounding.

5. The figure is difficult to read. Colored bar graphs have poor data to ink ratios. Try point and line figures with meaning groupings. Two figures, one for current IDU and one for former IDU would be better--and would have a better data to ink ratio. IDU is also better than PWID, as the figures describe an action (IDU) rather than a general group characteristic (PWID).

6. The authors rely upon self-reported HIV and HCV infection. The authors do not describe how this information was collected (e.g., were respondents asked about HIV and HCV infections directly or was a staged approach used--have you ever had a test for HIV? When was the last time you were tested for HIV? Has a health care provider ever told you that your test was positive?). There are a number of validated screening tools to ask this information. Were they used? Furthermore, the authors allude to results (presumably from the same project) indicating that "many of the current PWID had not undergone recent HIV or HCV screening." This further calls into question the accuracy of their results (while estimates of self-reported HIV and HCV infection are important, the actual seroprevalence rates are really what we want to know). The authors should also report HIV/HCV co-infection rates.

7. The limitations included are appropriate, notably that these results come from a single urban center that serves a marginalized community. The authors should address the additional limitations noted above (e.g., reliance on self-reported HIV and HCV infection, unclear impact of selection biases).

8. In the discussion, the authors state that "we believe the prevalence of PWID among urban ED patients is between 7% and 11%". The authors should temper their conclusion. Estimates vary based upon methods and study samples. Certainly the prevalence of injection drug use is significant and deserves attention, but accurate unbiased estimates of IDU among ED patients remain unknown.

9. On page 10, "We performed a cross-sectional survey study..." Suggest rewording to either "We performed a cross-sectional study..." or "We surveyed..." The terms cross-sectional and survey essentially mean the same thing.

10. On page 11, please define how you operationalized 'medically stable'. Non-medically (presumably) trained research assistants collected the data. How did they determine 'medical stability'?

6. PLOS authors have the option to publish the peer review history of their article (what does this mean?). If published, this will include your full peer review and any attached files.

Reviewer #1: No

---

## [Author Response · Author response to Decision Letter 0]

29 Apr 2020

Dear PLOS ONE Editorial board,

Thank you for your continued consideration of manuscript PONE-D-20-02701 "High Prevalence of Injection Drug Use and Blood-Borne Viral Infections In An Urban Emergency Department”. A revised version of the manuscript has been submitted along with this cover letter. 

We appreciate the reviewers’ excellent feedback and comments. Below we have included each reviewer comment, followed by our replies: 

Below we also address additional Editor and Reviewer comments. Our responses are in Bold and Italics. 

Additional Editor Comments (if provided):

I concur with the review regarding the weaknesses of the study: including the lack of information about people who did not take the survey, the survey instrument validity, the weaknesses of self-reported infection status and editing the figures. However, I think that the authors can address these critiques with a major revision, noting the more carefully the strengths and weaknesses of the study. For example, what are HIV and HCV testing rates in this area? If they are high, do they inspire confidence in the self-reported prevalence. Addressing the potential biases of the study are important and not over-interpreting the results.

We appreciate the constructive comments from the editorial team. We have added detailed information about the patients who did not complete the survey to the manuscript, addressed the limitations to the validity to the survey instrument, and discussed the weaknesses related to patient self-report of HIV/HCV status, as well as modifications to our figure. These are specifically addressed below when noted by the reviewer, and all changes to the manuscript are tracked in the re-submitted document. 

We have also added an additional figure for study enrollment and eligibility to delineate reasons patients were not able to participate, and have added an entire column to Table 1 with information about the entire ED census so that it can be compared to the study sample. We believe these editions will allow readers to interpret our findings with more context on potential methodologic biases. 

Specifically, with regards to this comment from the editor, in the Methods section of the manuscript, we have added information about both ED and community HIV/HCV. We have addressed the potential biases in this manuscript, with specific responses related to reviewer comments as noted below. 

Reviewers' comments:

Review Comments to the Author

Reviewer #1: The manuscript details results from a convenience sample survey of approximately 2,200 emergency department (ED) patients at a single urban ED (Highland) in Oakland, CA. The survey focuses on assessing injection drug use (IDU) history and self-reported HIV and hepatitis C (HCV) infections. The authors report high rates of IDU (and HIV and HCV) among their sample. The findings are relatively limited, including only a few demographic and visit characteristics. Within their findings, the note high reported IDU (e.g., 26% in respondents 25-34 years) and generally high rates of HIV (6%) and HCV (53%) infection.

The authors report a very high response rate--about 95%. The challenge is interpreting this number is that the authors do not describe the characteristics of 2,015 patients that they did not approach--and critically why not? Where these patients too ill? Intoxicated? Spoke a language other than English or Spanish? Discharged before the RA could approach them for participation? This information is critical to judge the validity of the remaining responses. The convenience sample design is practical, but it may have important implications on the findings. For example, if patients who inject drugs (or have HIV or HCV infection) have longer ED visits lengths of stay compared to their null counterparts, then the findings are subject to length biased sampling--and the results may be biased.

Thank you for this feedback. We have added to Table 1 the characteristics of the entire ED census during the study period. With this information presented side-by-side with the patients who completed the survey, we believe readers will have an objective assessment of any bias with survey administration. 

We have also added a new Figure 1, which shows the flow of patients who were eligible for survey administration, and the reasons patients were not approached for survey administration – of note, many patients had several reasons why they were not approached. 

The convenience sample design, single site selection, and lack of methodologic details limit the general applicability of their findings.

Yes, we agree these are limitations to our manuscript, and have expanded our limitations section to highlight these issues. We have also strengthened the methodologic details in order to address some of the important concerns mentioned by the reviewer. 

Specific comments:

1. Abstract: The authors state that they used a 'validated questionnaire' in the abstract. This is a bit of an overstatement. The authors used 2 validated question items from the NIDA Clinical Trials Network (CTN) Common Data Elements (CDE). This is a strength in their methods, but a general overstatement to imply that the questionnaire was validated (it would be fair to say that 2 items were validated). The authors do not provide their questionnaire for review. They do not describe the other survey data elements.

We have clarified the language in the abstract to say that our survey was based on validated surveys, rather than suggest that the entire questionnaire was validated. We have also provided the survey as an appendix.

2. The authors should clarify their use of 'Hispanic,' 'white,' and 'black'. In many contexts, Hispanic refers to ethnicity and white and black refer to race. I would presume that 'white' means 'non-Hispanic white' and 'black' means 'non-Hispanic black'. Again, if the questionnaire were provided as supplemental material, it would help gauge how the questions were asked.

We have clarified the race and ethnicity in Table 1; white refers to non-Hispanic white and black refers to non-Hispanic black. The survey is provided as an appendix. 

3. Were subjects compensated in any way? Were there any duplicate respondents (more than 1 ED visit during the study period)? If so, how were they managed?

We did not compensate patients for participation. Patients were not permitted to take the survey more than once, and during data analysis, duplicate patients were removed and only the initial survey was used in the data analysis. This information was added to the methods section of the revised manuscript. 

4. The statistical analysis is relatively simple, reporting percentages and absolute percentage differences between categories. The manuscript would be strengthened by a more robust analysis that included both absolute and relative differences (whether reporting as prevalence ratios or odds ratios). There are limited attempts to detect confounding or interaction (via limited stratification provided by text and not shown in tables). Sample size restrictions (of reported IDU, HIV, and HCV) limit the ability to adjust for confounding.

Thank you for this feedback. While our initial inclination was to display raw data with fewer statistical tests, we understand and agree with the rational for reporting a higher level of analysis. We have added absolute comparison between groups into Table 1, calculated as a difference in proportions with a 95% confidence interval. We have calculated unadjusted odds ratios to demonstrate relative differences between groups. With these additional analysis, we feel that our findings are more robust and convey more meaningful information for readers. 

We have also added to the results section summary comparison statistics, though refer to Table 1 to not repeat information in text form that is available in table form. 

5. The figure is difficult to read. Colored bar graphs have poor data to ink ratios. Try point and line figures with meaning groupings. Two figures, one for current IDU and one for former IDU would be better--and would have a better data to ink ratio. IDU is also better than PWID, as the figures describe an action (IDU) rather than a general group characteristic (PWID).

We appreciate the feedback on the figure. We have modified the figure (now Figure 2) into two figures (Figure 2a and Figure 2b), with one for current and the other for former injection drug use. We feel that the breakdown in each figure of non-Hispanic white and non-white by age group conveys meaningful information consistent with the public health dialogue surrounding injection drug use. The figure is also gray-scale for readability. 

6. The authors rely upon self-reported HIV and HCV infection. The authors do not describe how this information was collected (e.g., were respondents asked about HIV and HCV infections directly or was a staged approach used--have you ever had a test for HIV? When was the last time you were tested for HIV? Has a health care provider ever told you that your test was positive?). There are a number of validated screening tools to ask this information. Were they used? Furthermore, the authors allude to results (presumably from the same project) indicating that "many of the current PWID had not undergone recent HIV or HCV screening." This further calls into question the accuracy of their results (while estimates of self-reported HIV and HCV infection are important, the actual seroprevalence rates are really what we want to know). The authors should also report HIV/HCV co-infection rates.

We have added the survey as an appendix to clarify which questions were used for each datapoint. We also have added to the manuscript information about background rates of HIV and HCV testing in this population. While we do note that many patients had not had recent screening in our ED, there is widespread testing available for this community in our area – as well as our ED – and we have discussed this further in the manuscript in the background section. 

7. The limitations included are appropriate, notably that these results come from a single urban center that serves a marginalized community. The authors should address the additional limitations noted above (e.g., reliance on self-reported HIV and HCV infection, unclear impact of selection biases).

We have added an expanded discussion of the limitations of our findings, including those noted by both the reviewer and the editor. 

8. In the discussion, the authors state that "we believe the prevalence of PWID among urban ED patients is between 7% and 11%". The authors should temper their conclusion. Estimates vary based upon methods and study samples. Certainly the prevalence of injection drug use is significant and deserves attention, but accurate unbiased estimates of IDU among ED patients remain unknown.

Our language has been modified to temper our conclusions. 

9. On page 10, "We performed a cross-sectional survey study..." Suggest rewording to either "We performed a cross-sectional study..." or "We surveyed..." The terms cross-sectional and survey essentially mean the same thing.

We have changed this sentence to “We performed a cross-sectional study…”.

10. On page 11, please define how you operationalized 'medically stable'. Non-medically (presumably) trained research assistants collected the data. How did they determine 'medical stability'?

We have added to the methods how medical stability was assessed. 

We appreciate your continued consideration of this manuscript and hope that it meets the high publication standards of PLOS ONE. 

Sincerely,

Erik S. Anderson, MD

Corresponding Author

---

## [Editor Report · Decision Letter 1]

15 May 2020

High Prevalence Of Injection Drug Use And Blood-Borne Viral Infections Among Patients In An Urban Emergency Department

PONE-D-20-02701R1

Dear Dr. Anderson,

We are pleased to inform you that your manuscript has been judged scientifically suitable for publication and will be formally accepted for publication once it complies with all outstanding technical requirements.

With kind regards,

Kimberly Page, PhD, MPH

Academic Editor

PLOS ONE
---

## [Editor Report · Acceptance letter]

21 May 2020

PONE-D-20-02701R1 

High Prevalence Of Injection Drug Use And Blood-Borne Viral Infections Among Patients In An Urban Emergency Department 

Dear Dr. Anderson:

I am pleased to inform you that your manuscript has been deemed suitable for publication in PLOS ONE. Congratulations! Your manuscript is now with our production department. 

With kind regards,

on behalf of

Dr. Kimberly Page 

Academic Editor

PLOS ONE